# Theory Designed Strategies to Support Implementation of Genomics in Nephrology

**DOI:** 10.3390/genes13101919

**Published:** 2022-10-21

**Authors:** Arushi Kansal, Catherine Quinlan, Zornitza Stark, Peter G. Kerr, Andrew J. Mallett, Chandni Lakshmanan, Stephanie Best, Kushani Jayasinghe

**Affiliations:** 1Department of Nephrology, Monash Health, Melbourne 3168, Australia; 2Department of Medicine, Monash University, Melbourne 3168, Australia; 3Kidney Regeneration, Murdoch Children’s Research Institute, Melbourne 3052, Australia; 4Department of Paediatric Nephrology, Royal Children’s Hospital, Melbourne 3052, Australia; 5The KidGen Collaborative, Australian Genomics Health Alliance, Melbourne 3052, Australia; 6Department of Paediatrics, University of Melbourne, Melbourne 3052, Australia; 7Victorian Clinical Genetics Services, Murdoch Children’s Research Institute, Melbourne 3052, Australia; 8Australian Genomics Health Alliance, Melbourne 3052, Australia; 9Institute for Molecular Bioscience, The University of Queensland, Brisbane 4072, Australia; 10Department of Renal Medicine, Townsville University Hospital, Townsville 4817, Australia; 11College of Medicine and Dentistry, James Cook University, Townsville 4811, Australia; 12Northern Health, Melbourne 3076, Australia; 13Peter MacCallum Cancer Centre, Melbourne 3000, Australia; 14Victorian Comprehensive Cancer Centre, Melbourne 3050, Australia; 15Department of Oncology, University of Melbourne, Melbourne 3052, Australia; 16Melbourne Health, Melbourne 3050, Australia

**Keywords:** genomics, nephrology, theory designed strategies

## Abstract

(1) Background: Genomic testing is increasingly utilized as a clinical tool; however, its integration into nephrology remains limited. The purpose of this study was to identify barriers and prioritize interventions for the widespread implementation of genomics in nephrology. (2) Methods: Qualitative, semi-structured interviews were conducted with 25 Australian adult nephrologists to determine their perspectives on interventions and models of care to support implementation of genomics in nephrology. Interviews were guided by a validated theoretical framework for the implementation of genomic medicine—the Consolidated Framework of Implementation Research (CFIR). (3) Results: Nephrologists were from 18 hospitals, with 7 having a dedicated multidisciplinary kidney genetics service. Most practiced in the public healthcare system (n = 24), a large number were early-career (n = 13), and few had genomics experience (n = 4). The top three preferred interventions were increased funding, access to genomics champions, and education and training. Where interventions to barriers were not reported, we used the CFIR/Expert Recommendations for Implementing Change matching tool to generate theory-informed approaches. The preferred model of service delivery was a multidisciplinary kidney genetics clinic. (4) Conclusions: This study identified surmountable barriers and practical interventions for the implementation of genomics in nephrology, with multidisciplinary kidney genetics clinics identified as the preferred model of care. The integration of genomics education into nephrology training, secure funding for testing, and counselling along with the identification of genomics champions should be pursued by health services more broadly.

## 1. Introduction

Genomics is transforming clinical care of rare conditions through improved diagnosis and access to precision medicine [1,2,3] across many specialties, including neurology and oncology [4], whilst becoming an established first-line clinical tool in paediatrics [5,6]. A large number of genetic conditions present in nephrology practice, including but not limited to polycystic kidney disease, Alport syndrome, thrombotic microangiopathies, tubulointerstitial disease, and other glomerular diseases [2,7]. Genetic conditions contribute to a significant proportion of chronic kidney disease and are often underrecognized as a cause [8]. The potential benefits from routinely integrating genomic testing into nephrology are high, with multiple studies demonstrating high diagnostic and clinical utility [2,3,9], as well as cost-effectiveness [10]. Despite this, uptake has been limited, and genomic testing remains underutilised [11,12]. Across Australia there are some nephrology units that have access to an established kidney genetics service [3]. However, many centres have limited access to genomic input for their nephrology patients [3]. Recently, the Australian government provided reimbursement for genomic testing for the majority of patients with suspected monogenic kidney disease [13], which has the potential to significantly improve access to genetic testing within nephrology.

As we prepare to mainstream genomic testing into nephrology practice, there is paucity of data that evaluate the barriers to genomic implementation within nephrology [12]. Previous work in other specialty areas has explored barriers; however, few have used theory informed methods and focus, instead, on clinical knowledge and education [12,14,15]. Within nephrology, previous research assessed nephrologists’ attitudes and practices, as well as perceived barriers and interventions to the uptake of genomic medicine through a mixed methods electronic survey [16]. The findings demonstrated low uptake and confidence in genomic medicine by nephrology specialists, consistent with findings from studies of other specialists and primary care providers [17,18,19]. Most believed genomics to be useful in clinical care [2,16]. Numerous barriers to effective genomics implementation in nephrology were identified. Some of these barriers revolved around the working culture or ‘inner setting’ domains in the Consolidated Framework of Implementation Research [CFIR] [16,19]. The CFIR is a theory-informed conceptual framework designed to guide assessment of the implementation of complex health interventions. It draws on 19 implementation theories, models, and frameworks to structure the analysis of implementation barriers and interventions, and it comprises five domains: intervention source; outer setting; inner setting; characteristics of individuals; the process of implementation. Each domain has a range of sub-domains to provide more detailed analysis. To inform the development of implementation strategies, the CFIR can be used in conjunction with the Expert Recommendations for Implementation of Change [ERIC] [20]. The CFIR-ERIC matching tool provides a list of prioritised implementation strategies aligned with the CFIR constructs that can be used to tailor targeted approaches to support implementation. This tool was used when individuals offered barriers with no intervention, and this combined approach can offer potential theory-informed implementation strategies to support the uptake of a clinical intervention. 

Although a previous survey [16] provided important baseline data around implementation challenges of genomics in nephrology, further in-depth exploration of these barriers is required to inform future development of tailored solutions (CFIR/ERIC) to overcome these challenges. The application of qualitative methods permits the capture of a nuanced understanding of the implementation challenges facing nephrologists looking to use genomic testing in their practice. These data are essential prerequisites to the development of theory-informed interventions to support nephrologists. Therefore, using the CFIR to guide our study, we sought to (i) further examine the barriers nephrologists face when incorporating genomic testing into their practice, (ii) prioritize interventions to support nephrologists’ uptake of genomics, and (iii) identify preferred models of care.

## 2. Materials and Methods

### 2.1. Setting

This study included adult nephrologists who practiced in nephrology services in Australia. Multidisciplinary renal genetic clinics comprising nephrologists, clinical geneticists, and genetic counsellors operate in 15 public hospitals across Australia, and they serve each state/territory with some care provided privately [3]. Genomic testing is performed in a range of clinically accredited laboratories, which, in most cases, are sent locally. The types of tests requested are dependent on the clinical indication, with the majority of tests ordered being exome sequencing. Although there are differences among access to genetic services between states and territories, Australia has a public health care system, whereby at the time of this study, patients with clinically suspected Alport syndrome had access to federal funded genomic testing [21], whereas funded genomic testing for other kidney conditions was usually dependent on the availability of departmental funding at the individual health service and/or state funding level.

### 2.2. Individuals and Recruitment

To further explore barriers to implementing genomics in nephrology and investigate potential interventions from nephrologists’ perspectives, we purposively selected adult nephrologists from a diverse background to represent a variety of practices across Australia.

There were 22 public nephrology units that were emailed an invitation (K.J.) to participate in a qualitative semi-structured interview and/or nominate nephrologists within their department who were then invited to interview by email. Units were selected from categories, according to size (e.g., tertiary centre/number of dialysis patients serviced/location), to ensure the group adequately represented the distribution of the workforce. There were five groups of nephrologists interviewed to ensure all relevant individual groups were included: (1) Head of Unit, (2) Advanced Trainee, (3) Nephrologist Practicing Predominantly in a Public Hospital, (4) Nephrologist Practicing Predominantly in Private Practice, and (5) Nephrologist Involved with a Kidney Genetics Clinic. Individuals were from across five states and two territories for adequate geographical representation, including metropolitan and regional hospitals. Recruitment continued until data saturation.

### 2.3. Data Collection Tools and Procedure

We designed a cross-sectional qualitative study, building on formative research [16,22], using interviews (Appendix A) informed by the CFIR [19] to examine the barriers and interventions to the implementation of genomics in nephrology. 

Individuals who consented to the interview were emailed an invitation and a list of barriers and interventions for genomic implementation, which we referred to during the interview (Appendix A). Interviews were conducted one-on-one, via telephone, for a duration of thirty minutes on average. All were audio recorded with verbal individual consent, were fully transcribed verbatim, and were managed using NVivo [23]. Each transcript was de-identified using a unique identifier, with AT representing advanced trainee, CN representing consultant nephrologist, and HD representing head of department. Interviews were conducted by a trained Nephrology Advanced Trainee (A.K.) familiar with the interview to maintain consistency. 

The interview design guided by the CFIR, consisted of questions regarding individuals’ backgrounds, including qualifications, years of practice and organisation, view on barriers and interventions to the implementation of genomics in nephrology, and preferred models of service delivery for patients with kidney disease (Appendix A). The literature and previous research recognized known barriers to implementation and were shared with individuals. Individuals were asked to identify any additional barriers, implementation strategies, and to list top barriers and interventions for the implementation of genomics in nephrology. Individuals were asked to select their preferred model of service delivery from three options generated from the previous study [16] (Appendix A). Given the conversational style of the interview, further questions were individualised depending on each interviewee’s responses, and individuals were given the opportunity to provide additional comments and ask questions.

### 2.4. Data Analysis

Data were analysed and coded to identify themes and patterns using NVivo. The interviews were uploaded, and each was coded into broad categories based on CFIR. These categories included organisational view, current clinical model (and experience), interventions, and barriers. Barriers were then sub-categorised into general and pre-listed barriers. Interventions were sub-categorised as general, pre-listed interventional strategies, and future service models.

Using a deductive approach, we interrogated the interview data and identified barriers and implementation interventions. Using a modified CFIR coding table (Appendix A), we completed coding for the list of barriers. Transcripts were coded independently by three qualitative researchers (A.K., C.L., and K.J.), and discrepancies in coding—for example, where there was the potential for overlapping themes—were resolved through discussion to resolve uncertainty (K.J. and S.B.). Then, the barriers were grouped based on common themes and matched with interventions from the interview transcripts. Barriers without a suggested intervention were matched using the CFIR-ERIC matrix [24]. 

## 3. Results

First, we report on the demographics of the interviewees. Then, we present collated findings from the interviews about barriers, interventions, and models of service delivery for genomics implementation in nephrology.

### 3.1. Demographics of Interviewees

All individuals who responded to the email invitation were interviewed (n = 25, 100%), representing 18 out of 22 (82%) public nephrology departments that were invited to participate. Individual demographics are presented in Table 1. The majority of nephrologists were from Victoria or New South Wales, reflecting the most populous states in Australia with the greatest number of nephrology units. A large percentage of the clinicians were head of department in their respective hospitals (n = 9, 36%), and the remaining were either consultant nephrologists (n = 11, 44%) or advanced trainees (n = 5, 20%). More than 50% (n = 13) of individuals were early career nephrologists, and only 20% (n = 5) had been practicing for more than 20 years as a late career nephrologist. Only one individual had at least 50% of total clinical work in a private hospital, with the remaining 24 nephrologists predominantly working in the public system. There were few nephrologists (n = 4, 16%) with prior experience in renal genomics through direct involvement in a kidney genetics clinic or specific upskilling in genetic kidney disease. Nephrologists who were interviewed were collectively from 18 different hospitals. Among these, seven had access to an on-site multidisciplinary genetics clinic.

### 3.2. Barriers to Implementation of Genomic Testing in Nephrology

Common barriers to the implementation of genomic testing in nephrology were perceived lack of significant clinical impact of genomic testing, lack of knowledge, lack of funding for tests and clinical processes, and long waiting times. Table 2 provides exemplar quotes demonstrating the complete list of common barriers that were reported by individuals. Below, we discuss the most commonly reported barriers. 

#### 3.2.1. Perceived Impact of Results

Some nephrologists expressed doubt regarding the current knowledge and evidence-base for genomics and whether an established therapeutic role exists to justify genomic testing (CFIR code: Intervention Characteristics—Relative advantage). Given this is a relatively new field, this may highlight, again, an underlying lack of knowledge and exposure to the potential clinical utility of genomic testing.


*“There is a whole lot of uncertainty—firstly with indication, secondly when they get [it] what test should be ordered, and if they come back what the implications are for the patient, and how much can actually be done about that.”*
CN7

#### 3.2.2. Lack of Knowledge

One of the most common barriers, identified by nephrologists in this study, was a lack of knowledge (CFIR code: Inner setting—access to knowledge and information). Both lack of theoretical knowledge and skills to undertake the process of genomic testing (such as ordering the test, counselling, and results delivery), as well as referring to a genetics clinic, were highlighted by the interviewees. Most individuals (n = 21) did not have prior genomics experience.

Genomics was identified to be missing from the nephrology advanced training curriculum, as well as from general clinical exposure, at most hospital sites. Poor dissemination of knowledge was also recognized as contributing to the overall lack of knowledge. Some nephrologists also highlighted lack of knowledge and skills required to counsel patients. This was also compounded by lack of time for upskilling.


*“I think the genetics clinician is probably better in also counselling patients. I am not sure if nephrologists will have all the adequate training or experience to do all the counselling etc.”*
AT2

#### 3.2.3. Lack of Resources for Testing and Clinic Infrastructure

The significant costs of testing and consultations/resources required to implement this in clinical practice were raised as critical barriers to implementation in nephrology (CFIR code: Intervention Characteristics—costs). Until only recently, most renal genetic tests were not funded by the government, and without government-level support, there are limited other means to reduce the cost. 


*“If one needed to do that testing, it would usually need to go interstate and there is a current price tag of about $1500. The first question that the institution ask is where is the money going [to] come from to pay for that and often there is the justification that needs to be generated locally and it can be approved but there are always those difficulties.”*
HD5

In addition, many clinicians felt that there were insufficient resources available to support genomics in practice, with some clinicians also highlighting that funding is particularly inadequate for genomic testing, clinic infrastructure, and staffing (CFIR code: Inner setting—available resources). Insufficient government support was a key issue for most hospitals to maintain an active genetics service or those looking to establish a new service. Resources that were lacking, according to nephrologists, included support for test funding, staffing, and clinic infrastructure.

#### 3.2.4. Long Waiting Time for Consultation and Results

Amongst the nephrologists that we interviewed, many reported long waiting times as a significant barrier to utilising a genetics service and engaging patients (CFIR code: Inner setting—available resources). Long wait times applied for time from referral to review, as well as time to result return. 

In addition, the process itself is also time consuming, which can be difficult for both nephrologists and for patients. This was highlighted by some as not only an issue with the turn-around-time for a test but also for the review of results.


*“Often it is a year since the discussion was first raised with the patient about having genetic test. The testing process is certainly too protracted. Takes a long time to wait for the result to come.”*
CN10

### 3.3. Interventions to Support Use of Genomic Testing by Nephrologists

Several interventions were identified and highlighted by the clinicians in this study. The three key interventions that were most commonly prioritised to help integrate genomics into the field of nephrology included access to a genomics champion, funding, and education and training. The full list of barriers and interventions reported is listed in the Appendix A. 

#### 3.3.1. Informed Nephrologists as Genomics Champion 

A genomics champion was reported as being vital to promote the use of genomics in nephrology. This role was described by interviewees as suitable for a clinician with direct or indirect genomic involvement, who has some level of expertise and experience to provide advice, disseminate knowledge and information, provide a link to genetic services, and has a presence at unit meetings to identify patients that may benefit from genetics input. Among nephrologists interviewed with genomics experience, three out of four had played a key role in establishing a kidney genetics service at their respective hospitals. 


*“The most important ones really are a Genomics Champion, which we have but we really need that pushed along.”*
HD2

#### 3.3.2. Access to Testing—Including Funding

Increased funding was identified as a common preferred intervention for several of the barriers identified in this study, including long waiting times, lack of funding for testing and consultations, and lack of resources. However, the proposed source of funding support was not clear. Government support and funding is likely to be essential in implementing this intervention. The recent introduction of publicly funded testing is paramount to increasing accessibility to the tests currently available.


*“Funding support to get the test done to be able to pay for the cost of the genomic test which probably needs to happen at the central level. Medicare reimbursement type process but that doesn’t seem to exist.”*
HD5

#### 3.3.3. Greater Opportunities for Education and Training

In line with the barrier, ‘lack of knowledge’, education and training were frequently emphasized as major interventions required to assist with the integration of genomics into nephrology. Multiple strategies were suggested to implement this multifaceted intervention, including incorporating genetics training into nephrology physician training and increasing exposure at a trainee level.


*“It does need to be incorporated into our basic training—absolutely. Six-month rotation could be an option. It should be part of the 3-year course. You may get a minimum amount of time.”*
HD9

It was also suggested that genomics should feature more in conferences and educational meetings, including national nephrology conferences, seminars, and training courses attended by advanced trainees. 

Individuals reported that nephrologists who have already completed their training would also require a means to become abreast with genomics, so they will be able to comfortably integrate genomics into their everyday practice

To assist nephrologists with the referral and genomic testing process, one approach suggested was to develop educational materials that can be distributed amongst nephrology units for easy immediate access to this information. 


*“Have a clear form [for] what you need to do, what we can offer, cost, approximate waiting time and [whether] the result will be discussed by Geneticists and Counsellor or both.”*
CN6

### 3.4. Models of Service Delivery

Based on a previous study [16], nephrologists were provided with three options for models of service delivery for genomics, as listed in Table 3. Most individuals preferred the referral to a multidisciplinary kidney genetics clinic, which includes input from a geneticist, nephrologist, and genetic counsellor (n = 16, 64%). Less popular was allowing nephrologist autonomy in ordering genetic tests and requesting clinical genomics support as required (n = 3, 12%). The least preferred model was where the nephrologist refers onto clinical genetics (n = 2, 8%). Many highlighted a combination of models may be appropriate and more efficient depending on the clinical situation.

Other nephrologists highlighted the need to keep the responsibility of genomic investigations within the nephrology department. Many also commented that nephrologists would need to become more genomically literate and take responsibility to incorporate this into standard practice, in the future, to meet the demand.


*“I think it will. Now that genomic sequencing so cheap, it is increasingly incorporated into everything. I see that nephrologists requirements to have genetic skill will go up in the next 10 or 15 years or even before that. I would expect that nephrologists will be either running or being heavily involved with genetics supports is where it will probably head in the future.”*
CN9

## 4. Discussion

There is increasing recognition and appreciation of the value of genomic testing, in patients with kidney disease, amongst nephrologists [9,25,26]. With test funding becoming available in many public healthcare systems, including Australia, it is imperative that nephrologists incorporate genomics into routine practice. Previous research has examined aspects of genomics in nephrology, including the diagnostic utility of genomic testing [2,3,27], cost-effectiveness [10], and perceived barriers to the implementation [16,18]. A systematic review of studies which evaluated barriers and interventions of genomics implementation in other contexts demonstrated that few studies have assessed factors outside the individual and interpersonal levels [14]. In this study, we used a rigorous, theory-informed methodology that permitted a systematic in-depth analysis of the barriers and interventions identifying contextual, environmental and organisational barriers. This is the first study, to our knowledge, which explores nephrologists’ perspectives on preferred interventions and models of care.

Research in sub-specialities, other than nephrology, suggests that healthcare practitioners need a range of supports to facilitate incorporating genomics into their day to day practice, e.g., the need for effective education strategies [28,29], organizational support, and the importance of genetic counsellors to facilitate the implementation of genomics [30]. Despite advances in genomic testing and its demonstrated value, clinical genomic testing is often neglected as part of a routine nephrology assessment [11]. 

In this study, we sought to assess clinician-reported interventions that can help to assist with the integration of genomics, specifically, into nephrology. Our findings demonstrate that several effective interventions are perceived and recognised by nephrologists across Australia. In keeping with other studies, education and training were highlighted as major barriers to effective genomics implementation. Theoretical knowledge regarding genetics was reported to be deficient amongst trainees and nephrologists, with most individuals not having any prior genomics experience. Of note, some clinicians recognized the difficulty of further expanding the already crowded nephrology curriculum by incorporating genetics for trainees. This is consistent with the previous survey of nephrologists, as well as with other studies of medical subspecialists [16].

Interviewees highlighted a variety of interventions to improve education amongst nephrologists, including integrating genetics teaching into existing educational meetings and conferences. This alone however, will not enable effective use of genomics, as the practical challenges of test selection, and ordering process, result interpretation and counselling will likely still remain [16]. This finding highlights that there is a need to increase transparency about the testing and referral process to increase familiarity and comfort amongst nephrologists. Specific interventions highlighted in our study, which help overcome these practical challenges, include a variety of resources, such as educational materials for use in clinics (such as the development of guidelines and decision support tools), as well as local support, in the form of personnel, to help nephrologists incorporate genomics into their day-to-day practice. Interventions suggested by participants included a ‘help desk’, local access to a genetics counsellor/geneticist, and incorporating multidisciplinary meetings, which are already used in nephrology subspecialty areas, such as transplantation and dialysis [30].

Some clinicians questioned the role and need for genetics services in nephrology, given their doubt regarding overall utility. This barrier, CFIR coded as ‘Inner setting: tension for change’, was one of two barriers where clinicians themselves did not nominate a corresponding intervention. Therefore, a theory-informed CFIR-ERIC matched intervention was identified (conduct local needs assessment/facilitate relay of clinical data to providers). This intervention may help shift nephrologists’ perceptions of genomics and the place of genomics in clinical practice. Overall, these suggested interventions represent a need for cultural transformation to foster an appreciation for the value of genomics in nephrology.

Funding continues to play a central role in establishing and consolidating the use of genomics across nephrology. Without funding, individuals found it challenging to maintain current active services and struggled to set up genomic services and allow timely follow up. Increased funding was widely called for, by several individuals, as critical to support and facilitate the genomic testing and services provided. This is consistent with a previous study surveying nephrologists in the United States that also demonstrated cost was one of the most important barriers, as well as a lack of ease of use and poor availability of tests [18]. In addition to funding for testing, nephrologists reported a lack of overall resources to support and sustain the presence of genetics services across Australia. This includes access to genetics expertise, namely clinical geneticists and genetics counsellors. The lack of funding may also explain the long waiting times that were viewed as a barrier for both clinicians in two ways. First, delayed results can limit a clinician’s ability to provide care in a clinically relevant timeframe, and second, they can impact the utility and benefit of the test [31]. Some identified that long waiting time was also a barrier for patients, and whilst this may not be rectifiable with current turnover for results, it is possible to improve communication and address patient expectations. The barrier ‘Poor communication about timing of results’ was CFIR coded as Outer setting: Patient Needs and Resources and matched to the CFIR-ERIC intervention; ‘Involve patients/consumers and family members’.

### 4.1. Evolution and Adaptation of Genomics in Nephrology

Overall, nephrologists in this study recognised that genomics is constantly evolving [16]. The evolution of genomics in practice encompasses many aspects, including changes in sequencing techniques, legal requirements and data access/access to research, cost of technology, and indications and timing of sequencing in the diagnostic trajectory. Therefore, there is a clear need to develop adaptable interventions to support this evolution in practice. As an example, shortly after the interviews were conducted in our setting, the Australian government introduced publicly funded genomic testing for all patients with suspected monogenic kidney disease. Therefore, the frequently encountered barrier of cost for the genomic testing itself is no longer a barrier in our setting. Despite this dramatic improvement to access to testing, overcoming the test cost, itself, does not resolve many other issues which were frequently encountered by nephrologists in this study, namely lack of support for the resources, personnel, and infrastructure required to deal with the added complexities of genomic testing. 

Although short-term funding has enabled the establishment of kidney genetics services, there remains an overall lack of general funding to support and sustain rigorous changes in current practice and new service delivery models. Whilst the multidisciplinary team (MDT) model was the most preferred model of service delivery at present, some nephrologists also recognised potential limitations of this model and inefficiencies, as broader access to testing is required. This is a similar finding to other specialities, such as oncology, whereby novel models of care are being introduced to keep pace with the demand [32]. As demand increases, the MDT model for kidney genetics clinics may result in a bottleneck effect that prohibits timely and efficient outcomes for patients. Furthermore, with increased familiarity and experience with genomic testing, clinicians may prefer to initiate genomic testing and assessment before seeking MDT input. It is likely that the MDT model will need to evolve with competing workforce constraints and the need to mainstream some aspects of genomics in nephrology [33].

### 4.2. Strengths and Limitations of This Study

We adopted a theory-informed approach to this study. The use of the CFIR has been well established in genomics [33,34] and can provide rigor and structure to the assessment of implementation barriers and interventions in context. Vitally, the CFIR constructs have been matched with the ERIC framework [20].

All interviews were conducted by one interviewer (A.K.), hence ensuring consistency in questioning and conversation. The findings build upon previous research [3,16,22] and as a novel study, we have, for the first time, documented potential implementation strategies that can be further explored and trialled within nephrology in the Australian healthcare system.

Our study was conducted within a publicly funded healthcare system and, therefore, results may be difficult to generalise to different health structures, though they may provide useful insight for others based in, primarily, public health systems. We interviewed adult nephrologists from different hospitals across Australia; however, the majority were from New South Wales and Victoria, and the majority were from tertiary metropolitan hospitals. Although advanced trainees, heads of department, and nephrologists with genetics experience were interviewed, we had poor representation from the private health sector with only one nephrologist, thus introducing potential for selection bias. Further research is required to identify the nuances that may apply to private healthcare, as well as regional and rural sites. 

### 4.3. Future Directions

Future research needs to evaluate implementation strategies for genomic interventions using an implementation science framework to assess impact and help inform the scalability of effective strategies to diverse populations and settings [33]. Additionally, the best model of service delivery may need revision in the future, as the currently preferred MDT model may no longer remain robust as genetics becomes increasingly part of routine nephrology assessment.

## 5. Conclusions

Our study has highlighted an array of interventions that are commonly viewed by nephrologists as being potentially effective in the implementation of genomics in nephrology. This reinforces and deepens previous findings to inform future implementation interventions. There is an apparent need for more active discussion and collaboration to accelerate this implementation. The most common interventions that were prioritized by nephrologists include genomics champions, increasing knowledge via education and training, and access to funding. Further studies should focus on implementing and evaluating outcomes of these interventions in the field of nephrology, as well as developing models of care that are able to evolve to meet the increased demand for testing that is likely to develop in the future.

## Figures and Tables

**Table 1 genes-13-01919-t001:** Demographic profile of interviewees.

Category		n, (%)
State of practice	Victoria	8 (32)
	Queensland	3 (12)
	South Australia	3 (12)
	New South Wales	6 (24)
	Australian Capital Territory	2 (8)
	Tasmania	2 (8)
	Western Australia	0 (0)
	Northern Territory	1 (4)
Position	Advanced trainee	5 (20)
	Consultant nephrologist *	11 (44)
	Head of Department	9 (36)
Career Stage	Early career (0–10 years)	13 (52)
	Mid-career (11–20 years)	7 (28)
	Late career (20+ years)	5 (20)
Predominant sector > 50%	Public	24 (96)
	Private	1 (4)
	Prior experience	4 (16)
Genomics Clinic Experience	No prior experience	21 (84)

* Consultant nephrologist includes all nephrologists with Fellowship of Royal Australasian of College of Physicians.

**Table 2 genes-13-01919-t002:** Common barriers coded using the CFIR with exemplar quotes and matching interventions.

CFIR Code	Barrier	Quote from Text	Matched Intervention from the Interview Transcripts	Quote from Text
Inner setting: Available resources	Long waiting time for clinic	Patients have reported back saying there is a 3–6 month wait for an opinion. CN2	Funding for testing and clinic	Funding support for the test and consultation is also very important. CN1
	Lack of resources	I would say that we are fairly dysfunctional in the genetics component, largely perhaps due to a combination of personnel, infrastructure and funding being the reasons. HD1	Funding for genomics service	Funding support—that is going to become critical. HD6
	Lack of genetics expertise	Even if there is funding, it is very hard to recruit people to work in XX and I don’t think any clinical geneticist is ever going to set foot in XX to work. CN8	Easier access to local genetics expertise	Easy access locally. It would be really good even if we had a genetics counsellor locally, so once we get some results back to talk through what they mean with the patients, and also help us interpret what the test is might be really useful. CN5
Intervention characteristics: Adaptability	Long turn-around time for results	…the tests take a long time to come back, so that’s one common feedback from the patients as well. CN2	Funding for testing and clinic	Funding support for the test and consultation is also very important. CN1
Outer setting: Patient Needs and resources	Poor communication about timing of results	It is not a quick turnaround test and the communication around that probably has need for improvement. HD6	Involve patients/consumers and family members ^1^	
Intervention characteristics: Cost	Lack of funding overall	They probably do need more funding and resources to implement some of their models of care. CN10	Funding for testing and clinic	Funding support for the test and consultation is also very important. CN1
	Lack of funding for testing	There is a very small amount of money that the department will allocate to genetic tests. CN8	Funding for testing	Funding support to get the test done to be able to pay for the cost of the genomic test which probably needs to happen at the central level. Medicare reimbursement type process but that doesn’t seem to exist. HD5
	Lack of funding for clinic	It is not a funded clinic and is not incorporated in usual hospital outpatients and is such a small component. HD1	Funding for testing	Funding support to get the test done to be able to pay for the cost of the genomic test which probably needs to happen at the central level. Medicare reimbursement type process but that doesn’t seem to exist. HD5
	Lack of funding for staff	There are potential people who are extremely suited for the job but we don’t have enough money to create another position. HD4	Funding for clinic	Funding support for the test and consultation is also very important. CN1
Characteristics of individual: Knowledge and beliefs about the intervention	Perceived impact of results	Genetics is always going to be hard to give a patient a definite answer on, but presumably that will improve as we collect more data. CN1	Genomics Champion	The most important ones really are a Genomics Champion, which we have but we really need that pushed along. HD2
Process: Champions	Lack of interest	Whereas with the genetics clinic, I feel like there is probably going to be a lot of barriers to setting one up, particularly if there is not a lot of interest from the nephrology department. AT4	Genomics Champion	The most important ones really are a Genomics Champion, which we have but we really need that pushed along. HD2
Inner setting: Access to knowledge and information	Lack of time for learning	People are already time-poor, the average age of a nephrology trainee is getting older and you have competing priorities on time, so how are you going to put in a completely new area, superimposed on what is already a very grounded curriculum. HD9	Genetics training for nephrology trainees as a rotation	It does need to incorporated into our basic training—absolutely. Six-month rotation could be an option. HD9
	Lack of theoretical knowledge	I think that needs a lot more discussion and a lot more presence at national meetings and ANZSN. HD2	Educational meetings	Some incorporation of more education. Update course and kidney school. CN3
	Lack of knowledge about process	I think executive summary, very early on we all think it is going to be important in the not too distant future but currently most people are unsure what tests are out there, who we should be referring to, and exactly how we should doing it—in terms of the logistics of including [the] right paperwork or the right person. CN6	Develop educational materials	Have a clear form, what you need to do, what we can offer, cost, approximate waiting time and [if] the result will be discussed by Geneticists and Counsellor or both. CN6
Inner setting: Tension for change	Perception of need for service	I don’t think there is enough business and scope to have a whole single nephrology genetics clinic, certainly on a weekly basis or perhaps even a monthly basis. HD1	Conduct local needs assessment/facilitate relay of clinical data to providers ^1^	

^1^ CFIR-ERIC theory informed.

**Table 3 genes-13-01919-t003:** Model of service delivery.

Model	n, (%)	Quote from Text
Nephrologist refers to multidisciplinary renal genetics’ clinic	16 (64)	I like the multidisciplinary clinic. So it’s a one-stop-shop for the patient. The diagnostics, counselling and the support at the same time, together with the nephrologist referring the patient as ongoing care and implementing. HD9
Nephrologist orders test and returns result with clinical genetics support as needed	3 (12)	Being able to order tests yourself, you can overcome the barrier of waiting lists and then liaise directly with the Renal Geneticist and things get done before the patient be seen in the clinic. …So if we can get a start on some of the investigations and tests before going to the Renal Genetics clinic, it may also make that a bit more efficient. CN2
Nephrologist refers to clinical genetics	2 (8)	I would prefer the third model. The reason being I worked in cancer care and maternity services. In those two services, they have an external genetic service and I think their model works quite well and their geneticists are linked in with their services. Again in neither of those services, the genetic referrals are not common but get enough volume to have an established relation and that worked quite well. CN11
Combination of two or more models	4 (16)	Combination of the first two is my preferred model. I am comfortable ordering a test for Tuberous Sclerosis for example working in the area but I won’t order tests for other panels that I am not comfortable with. To use a multidisciplinary renal clinic with a geneticist and a nephrologist it would add depth to the clinic. CN10

## Data Availability

Not applicable.

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
