# Peer review of "Theory Designed Strategies to Support Implementation of Genomics in Nephrology"

_genes, 2022, doi:10.3390/genes13101919_

Round 1

Reviewer 1 Report

Dear Authors,

The manuscript described a study using ERIC tool matched to CFIR impressions of interview data obtained from adult nephrology providers in Australia, and how the tools are used to identify intervenable barriers to the application of Genomic testing in nephrology.  This is an interesting and great topic for discussion and I appreciate that the investigators are advocating for Genomic testing in nephrology

The major problem of this work is the measurements are based on the impressions using a generalized CFIR tool, which limited the dissection of barriers that incite realistic interventions and quality improvement. The authors mainly focused on resolving the barriers to the application of Genomic testing. However, the barriers are not necessarily barriers, genomic tools, if they are to improve the practice in the subspecialty, need to be used appropriately, with clear indication, consent, pre-diagnostic prediction. The authors may consider adding the justification to correct the barriers.

Anther problem is, the barriers and interventions are lack of details, except quoting the text of the interviews. If there is a high lack of knowledge in the subjects that are being interviewed, it will be more valuable to interview experts in clinical genetics to expand the candidate barriers and interventions to be included in the study.

Below are my concerns and suggestions

 Table 1 listed only 4 providers who had had prior experience in renal genetics. However, in abstract, “seven” having a dedicated multidisciplinary kidney genetics service? Please clarify how the 3 providers have a dedicated kidney genetics service without experience.

2.       There are many overlaps in the barriers described in both the main text and the table 2.

I suggest summarizing them in a more organized, less overlapping, and more precise manner.

3.       Monogenic kidney disease spectrum that might warrant genomic testing are quite different between adult and pediatric renal patients, please specify the spectrum of genetic kidney disease for which you are trying to identify the barriers in the adult nephrologists. For instance, TSC, ADPKD, assess post-transplant recurrence in patients with family history, FSGS, aHUS…...

4.       The description of data analysis: The measurements are largely descriptive and subjective but could be improved.

For instance, from a 0-10 scale, the education level of renal genomics (something like: 6 being having participated in genomic workshop; 8 being having participated in renal genomic workshop; 9 being having participated or proposed a genomic project)

In the CFIR supplementary table A1, Relative Advantage, I did not see the part regarding “Stakeholders’ perception of the advantage of implementing genomics in nephrology versus an alternative solution” was described or studied.  This is important to help you justify why you want to resolve the barrier to applying genomic tests.

5.       While reviewing the barriers noticed by the providers, some of them are quite fair. When the authors are labeling some of them as barriers, would you also consider exploring what made them barriers…for instance, “Some nephrologists expressed doubt regarding current knowledge and evidence-base for genomics and whether established therapeutic role exists to justify genomic testing” this may highlight again an underlying lack of knowledge…the doubt is very fair and can’t be only categorized in lack of knowledge. Maybe should include barriers like “lack of pre-test counseling and consent for both providers and patients”.

3.2.2 knowledge: The barriers the authors tried to score are extremely vague. Suggest to further subcategorized into more objective ways for a better scoring, for example

From a 0-10 scale, the education level of renal genomics (something like: 6 being having participated in genomic workshop; 8 being having participated in renal genomic workshop; 9 being having participated or proposed a genomic project)

From a 0-10 scale, the practical level of renal genomics (6 being having ordered a genetic test, 7 being having referred patients to genomic clinic or multidisciplinary TSC clinic…8 being purposefully obtained genomic data to guide targetable rare renal disease like primary hyperoxaluria type I).

You don’t have to do it as I suggested but can consider using scales to help interpret your findings for this and future studies.

3.2.3. resource: Do they have a dedicated or shared genetic counselor to facilitate the pre-test and post-test counseling, tests to choose, obtaining consent?

Lack of funding for tests: please specify, were the costs denied by insurance company, are there opportunities to share the costs with research funding, insurance company, social security, or community coverages; are there chanced that third party testing agents or laboratories that can significantly reduce the costs.

3.2.4. Re long waiting time, recommend specifying what tests are offered, which labs are offering these tests (commercial, governmental labs, academic labs, etc.). If you want to emphasize long waiting time, please avoid including other confounding factors in the same quote “for instance, funding”.

3.3.1:  Re Genomic champion, probably more valuable to describe the structure of a renal clinic that currently offering genomic test and what the barriers are. For instance, do you have IRB office, structured way of developing a consent, genetic counselors, and nurses, and a current practice guideline or some guideline that is to be developed. These probably are more important than relying on a champion.

3.3.2. re funding, it will be more valuable to discuss what realistic intervention you can consider, in addition to increasing or finding the funding”. For instance, what about reduced cost by choosing the most targeted tests, shared cost with insurance company or other source, reduced cost by pursuing third party companies who can provide a better cost/effect ratio

Table 2:

Quote from the text could be shorter to emphasizing key words, otherwise the readers might feel exhausted after reading a couple of lines.

Domains and variables could be organized in a less redundant manner. For instance, anything focusing on intervention: costs can be organized as subcategories under one Intervention characteristics.

Re available of resource, Barrier is more than long waiting time. Please consider including others like funding recourses, practice site and providers that can offer the evaluation, etc. The intervention is not matched to the barrier, please match accordingly.

Re adaptability, Barrier is more than long turnaround time for results. Please including something like testing laboratories, the intervention is not matched to the barrier, please match accordingly.

Thank you 

Re Patient Needs and resources, Barrier is more than Poor communication about timing of results. Please consider dedicated counselors, pre and post testing counseling. And match more interventions to more detailed barriers.

Re lack of funding, barrier could include resources of shared cost, different tests offered commercially that could reduce the cost, understanding of cost/effect relationship compared to alternatives.

4.2. and 4.3. limitations and future studies

Regarding the study, do you have data or a plan of comparing adult nephrologist with pediatric nephrologist, or with adult nephrologists with another adult subspecialty where genetic tests are applied, or with the practice from another country with a more structured concept, or compare the practice with practice guideline in adult nephrology in Australia, or in international society of nephrology.

Reviewer 2 Report

1. Section on materials and methods, individuals and recruitment - in my opinion more detailed information concerning selection methods is necessary i.e. how many units/private/public had been approached, what was the response rate.  The authors only state that:  All individuals who responded to the email invitation were interviewed (n=25,100%)

2. In my opinion lenghty citations from interviewees responses are unnecessary and compromise the clarity of the paper. The authors should consider rather more synthetic presentation of data.

3. The authors  attempted to describe the strengths and limitations of the study in section 4.2, however I would suggest more critical approach.  The authors state that they had interviewed a representative group of adult nephrologists from different hospitals across Australia etc. however as mentioned in p. 1 there has been limited data provided concerning study group. Further:  20/25 (80%) were Heads of Department/Consultants (what about other practisng nephrologists ?) 56% of all interviewed where from only two states and only 1/25 (4%) from private sector.

Author Response

Reviewer 2 Comments

Point 1: Section on materials and methods, individuals and recruitment - in my opinion more detailed information concerning selection methods is necessary i.e. how many units/private/public had been approached, what was the response rate.  The authors only state that:  All individuals who responded to the email invitation were interviewed (n=25,100%)

Response 1: Thank you. 22 units (all public) had been approached and response rate was 82%. This information has been added in the manuscript (Section 2.2 & 3.1, Page 3).

Point 2:  In my opinion lenghty citations from interviewees responses are unnecessary and compromise the clarity of the paper. The authors should consider rather more synthetic presentation of data.

Response 2: Thank you for your comments. Reporting qualitative data succinctly is challenging. We have reviewed the interviewee responses included in the paper and rationalized and synthesized the results where possible.

Point 3: The authors attempted to describe the strengths and limitations of the study in section 4.2, however I would suggest more critical approach.  The authors state that they had interviewed a representative group of adult nephrologists from different hospitals across Australia etc. however as mentioned in p. 1 there has been limited data provided concerning study group. Further:  20/25 (80%) were Heads of Department/Consultants (what about other practisng nephrologists ?) 56% of all interviewed where from only two states and only 1/25 (4%) from private sector.

Response 3: We have revised the materials and methods to include the number of hospitals emailed and response rate. We have used the term 'consultants' as one that encompasses all practising nephrologists in Australia which we have now noted in Table 1. The limitations section has been revised to include comments on states and private sector. The Australian population is unevenly distributed geographically, and the healthcare system is largely public: the participation rates reflect this.

Round 2

Reviewer 1 Report

Dear Authors, 

I am glad to see my suggestions are helpful.

The revised manuscript addressed the concerns. Therefore, I will recommend acceptance for publication. 

Thank you 

Sincerely

Hua Sun, MD. PhD

University of Iowa